# Aircraft Autonomous Separation Assurance Based on Cooperative Game Theory

**Xinmin Tang** [1,2,*], **Xiaona Lu** [1] and **Pengcheng Zheng** [1]

1    Civil Aviation College, Nanjing University of Aeronautics and Astronautics, Nanjing 211106, China; luxiaona@nuaa.edu.cn (X.L.); zpc@nuaa.edu.cn (P.Z.)
2    College of Transportation Science & Engineering, Civil Aviation University of China, Tianjin 300300, China
*    Correspondence: tangxinmin@nuaa.edu.cn

**Abstract:** Transferring part of the separation assurance responsibilities from air traffic controllers to pilots during en route phases of flight can reduce the controllers' workload while ensuring operational safety and improving operational efficiency in the airspace. For this new generation of distributed air traffic management mode, firstly use the conflict detection algorithm to determine whether a potential conflict exists between two aircraft, introduce cooperative game theory to autonomous separation assurance model for horizontal cross-conflict in a static wind field by forming a coalition of all aircraft involved in the potential conflict. The convex combination of minimum yaw angle and maneuver flight time is used as the strategic gain of the aircraft, and the welfare function of the coalition is maximized by changing the behavioral strategy of the aircraft. Finally, a horizontal cross-conflict scenario is set up for simulation experiments and compared with a centralized separation assurance strategy. The simulation results show the effectiveness of cooperative game theory, which is applied in distributed autonomous separation assurance.

**Keywords:** air traffic management; autonomous separation assurance; cooperative game theory; conflict resolution

## 1. Introduction

The concepts of free flight, autonomous aircraft separation assurance, distributed air traffic management, etc., were proposed by Europe and NASA in the late 20th century for the shortcomings of the traditional air traffic management mode. These new operational concepts are aimed to use the advanced airborne equipment to share part of the separation assurance responsibilities, thereby reducing controllers' workload while making use of airspace more efficiently and improving the capacity and flow of the airspace [1,2].

Under operational conditions of partially authorized separation assurance responsibilities, compared to the centralized calculation of separation assurance solutions on the ground, the distributed calculation by aircraft can significantly reduce the complexity of the calculation, improve the efficiency and real-time of separation assurance, and the aircraft can select the preferred route, altitude, and speed, effectively reducing fuel consumption and flight delays. At present, the own ship can obtain the status information of aircraft around its own ship via ADS-B IN, but it is difficult to obtain the intention information and the separation assurance strategy adopted by aircraft around the own ship when they encounter separation loss, which may cause the failure of the separation assurance control law calculated by own ship, and then lead to the separation between two or more aircraft not meeting the safety requirements.

In the traditional own ship maneuver separation assurance, after the own ship detects a conflicting aircraft with which there is a potential separation loss, it is assumed that the conflicting aircraft continues to fly along the nominal trajectory, and the way to avoid conflict between the two aircraft is to calculate and execute own ship's separation assurance control law, which means the conflicting aircraft does not change its trajectory and only

own ship performs maneuvering flights. This approach is suitable for mixed airspace in which aircraft are significantly different in airborne equipment performance. In the mixed airspace, the aircraft with more advanced airborne equipment takes the responsibility of separation assurance, which usually means that it needs to perform large maneuvering flights to ensure operational safety, and the comfort, economy, and safety of this aircraft will be greatly affected.

In order to ensure safety under operational conditions of partially authorized separation assurance responsibilities, some scholars at home and abroad have introduced game theory into aircraft operation and conflict resolution:

Claire Tomlin et al. [3,4] introduced non-cooperative game theory to multi-aircraft conflict in a free-flight environment by developing a corresponding conflict resolution method for each aircraft, in which each aircraft develops a resolution strategy for the worst-case scenario that may occur to the other aircraft. Yang Min et al. [5] used cooperative game theory to study the conflict problem of high-speed ramp convergence by forming a coalition of vehicles involved in the conflict. Cheng Ying et al. [6] proposed a conflict resolution method for non-signalized intersections based on multi-vehicle cooperative optimization and solved the problem of unclear road right of way of traditional autonomous vehicles at non-signalized intersections. Erokhin, V et al. [7] suggested an approach based on game theory to solve the problem of bi-criteria control and optimization of an aircraft flight trajectory using the data of satellite navigation systems. Xu K et al. [8] first built a cooperative game model that makes each player consider the preferences of the other players through a proposed priority ranking mechanism and then used a probabilistic prediction model to describe the resolution of the game conflict, which can eventually resolve the flight conflict and satisfy individual preferences at the same time. Sang, G.P et al. [9] raised a new trajectory negotiation mechanism which is formulated as n-player, finite strategy game. The objective of each aircraft in the game is to minimize the cost of deviating from its desired trajectory, while the objective of the controller is to provide clearances that ensure fairness while preventing conflicts. Li, T et al. [10] examined the benefits of reducing the separation standards between flights. Air traffic control agencies conduct benefit–cost analyses of adopting new/improved technologies, airlines develop strategies to best utilize satellite services, and satellite service providers design fee-for-service programs and conduct market analyses. Baspinar, B et al. [11] focused on modeling air-to-air operations through an optimization-based control and game theory approach. With the help of game theory, a battle between two aircraft is transformed into an optimization problem which is solved with a moving time horizon scheme to produce the best strategy for the aircraft in air combat. Zhang, B.C et al. [12] analyzed airport congestion under the Stackelberg game by considering flight price discrimination between different types of aircraft and introduced a method to determine the specific total number of flights. Garcia, E.A et al. [13] studied a coastline or boundary which was under attack by two aircraft, with the maritime boundary being guarded by the faster defender. The two aircraft cooperate and try to minimize their total distance to the boundary as each aircraft is intercepted by the defender. Sharma, M.G et al. [14] considered cooperative arrangements for the supply of spare parts between two or more airlines, modeled possible alliances in an aircraft spares supply scenario, and used core concepts from cooperative game theory to investigate stable outcomes.

The above latest references applied non-cooperative and cooperative game theory to the conflict problem of ground vehicles and air traffic, which not only efficiently solved the traditional conflict resolution problem but also considered the decisions generated by other participants, avoiding the current conflict resolution strategy leading to conflict with other participants and achieved the overall optimization of the system eventually.

Therefore, in this paper, we consider that all aircraft in the airspace have autonomous separation assurance capability and use a cooperative game to solve the distributed autonomous separation control. After detecting potential separation loss, forming a coalition of the involved aircraft to jointly negotiate a suitable separation assurance control law, constructing an autonomous separation assurance model in a horizontal cross-conflict scenario, and using a convex combination of minimum yaw angle and maneuver flight time as the

aircraft's strategy gain. The welfare function of the alliance is maximized by changing the behavioral strategy of the aircraft. This autonomous separation assurance method is collaborative and distributed, and the aircraft does not need significant maneuver flights to avoid safety separation loss, so it also improves operational safety.

## 2. Instantaneous Wind Field Model

### 2.1. Aircraft Speed Triangle

Aircraft are often affected by upper wind during the flight phase of the route, and the uncertainty of the wind vector makes the difference between the airspeed vector used by the aircraft and the ground speed vector used by the ground controller, and the real-time ground speed vector of the aircraft can be solved based on the airspeed vector and the wind speed vector [15], as shown in Figure 1. The notations and their corresponding concepts in Figure 1 are shown in Table 1.

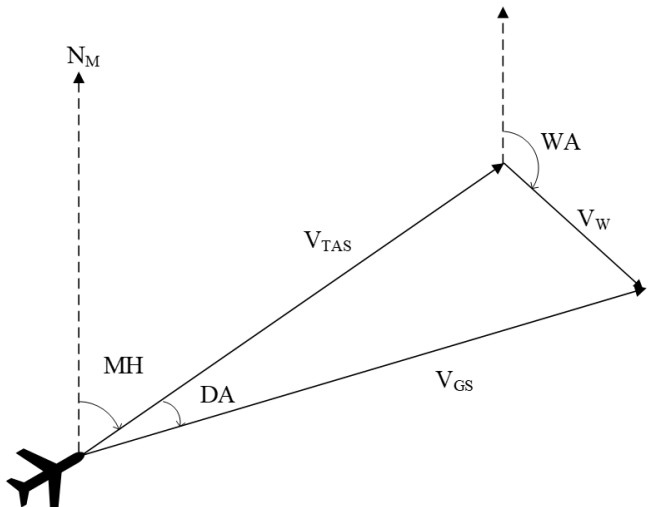

**Figure 1.** Aircraft speed triangle.

**Table 1.** Notations and their corresponding concepts of Figure 1.

| Notations | Concepts |
| --- | --- |
| $V_{TAS}$ | vacuum velocity |
| $MH$ | magnetic heading |
| $DA$ | drift angle |
| $WA$ | wind angle |
| $V_W$ | wind speed |
| $V_{GS}$ | ground speed |
| $N_M$ | magnetic north |

From the aircraft speed triangle, it follows that:

$$\overrightarrow{V_{GS}} = \overrightarrow{V_{TAS}} + \overrightarrow{V_W} \tag{1}$$

then the aircraft kinematic equation in the ground inertial reference coordinate system is

$$\begin{cases} \dot{x} = v_1 \times \cos\varphi + \omega_1 = v_2 \times \cos\theta \\ \dot{y} = v_1 \times \sin\varphi + \omega_2 = v_2 \times \sin\theta \end{cases} \tag{2}$$

Variables and corresponding concepts of above equation are shown in Table 2.

**Table 2.** Variables and corresponding concepts of kinematic equation of the aircraft.

| Variables | Concepts |
|---|---|
| $v_1$ | vacuum velocity of the aircraft |
| $v_2$ | ground speed of the aircraft |
| $\varphi$ | heading angle of the aircraft |
| $\theta$ | track angle of the aircraft |
| $x, y$ | lateral and longitudinal positions of the aircraft |
| $\dot{x}, \dot{y}$ | lateral and longitudinal components of the aircraft velocity |
| $\omega_1, \omega_2$ | lateral and longitudinal components of the wind speed |

*2.2. Instantaneous Wind Field Model*

According to the established coordinate system, the wind vector can be decomposed into two components parallel to and perpendicular to the longitudinal axis of the aircraft. The composition of the wind can be divided into two components, the forecast wind vector and the forecast error in the actual system [16,17]. The forecast wind is the wind field information of the region measured by the meteorological department through weather radar, aircraft reports, etc.; the wind error is not considered in this paper. The wind field information where the aircraft is located is shown in Figure 2.

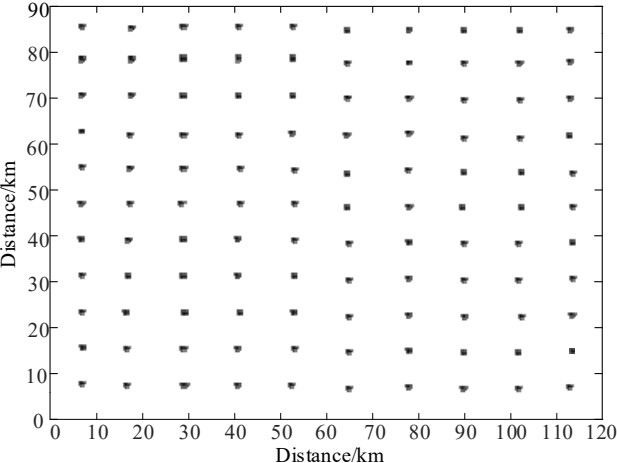

**Figure 2.** Instantaneous wind field information at the altitude of the aircraft.

## 3. Conflict Detection Algorithm

According to the Air Traffic Control (ATC) regulations issued by the Civil Aviation Administration of China (CAAC), if the distance between two aircraft is less than 10 km, it is considered that a potential conflict exists between two aircraft, and the corresponding conflict resolution strategy needs to be implemented immediately in order to ensure flight safety.

In the model, the aircraft is approximated as a mass point, and its attitude is ignored. According to Equation (2), under the condition that the deflection angle is small $\varphi \approx \theta$ then its kinematic equation in the ground inertial reference coordinate system is

$$\begin{cases} \dot{x} = v_1 \times \cos\varphi + \omega_1 = v_2 \times \cos\theta \approx v_2 \times \cos\varphi \\ \dot{y} = v_1 \times \sin\varphi + \omega_2 = v_2 \times \sin\theta \approx v_2 \times \sin\varphi \end{cases} \tag{3}$$

The initial state of aircraft $A$ is $Z_{a0} = (x_a, y_a, \varphi_a)$, the ground velocity vector is $\dot{Z}_a = (v_a \cos v_a, v_a \sin v_a)^T$, The initial state of aircraft $B$ is $Z_{b0} = (x_b, y_b, \varphi_b)$, the ground velocity vector is $\dot{Z}_b = (v_b \cos v_b, v_b \sin v_b)^T$. $x_a, y_a, \varphi_a, v_a$ are the lateral position, longitudinal position, heading angle and ground speed of aircraft $A$, respectively. $x_b, y_b, \varphi_b, v_b$ are the lateral position, longitudinal position, heading angle and ground speed of aircraft $B$,

respectively. In the plane right angle coordinate system, the aircraft $B$ relative to the aircraft $A$ [18]:

$$\begin{cases} Z_r = \begin{bmatrix} x_b - x_a \\ y_b - y_a \end{bmatrix} \\ \varphi_r = \varphi_b - \varphi_b \end{cases} \tag{4}$$

Under the background of the continuous expansion of the radar control area of civil aviation in China, the separation between multiple aircraft is also further reduced. The minimum horizontal separation of civil aviation in China at low and medium altitudes is reduced to 6 km, and the minimum horizontal separation at high altitudes is shortened to 10 km. Therefore, this paper simplifies the safety separation accordingly and agrees that the aircraft is cruising at a high altitude, and the minimum horizontal separation is proposed to be 10 km.

$\| Z_r \| = \sqrt{(x_b - x_a)^2 + (y_b - y_a)^2}$, If $\| Z_r \| \leq 10$ km, a potential conflict between two aircraft will be detected at the same altitude. The relative motion method can be used to ensure the minimum safety separation mentioned above. The conflict detection schematic diagram is shown in Figure 3.

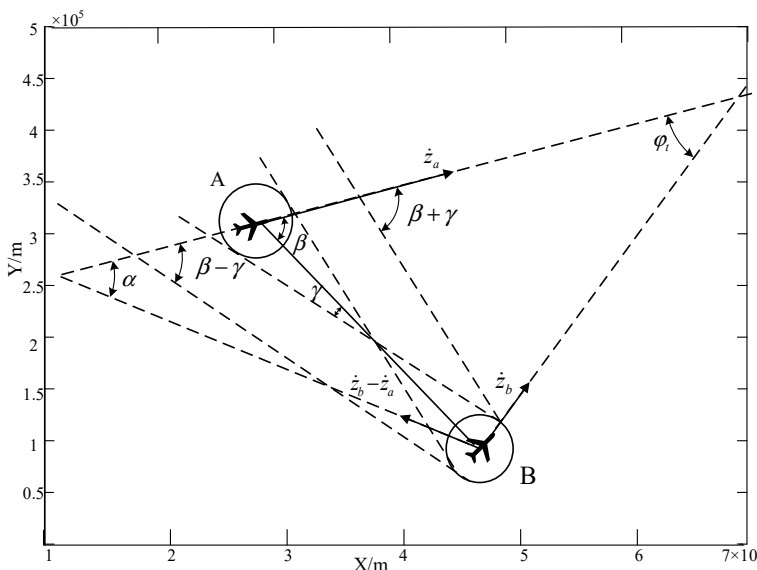

**Figure 3.** Schematic diagram of two-aircraft conflict detection.

The radius of the protection area for each aircraft is 5 km, and two straight lines are tangent to aircraft $B$ and the relative velocity vector $\dot{Z}_b - \dot{Z}_a$ is parallel to the above two straight lines. The flight corridor in which aircraft $B$ moves along the direction of aircraft $A$ is also formed by the enclosed area, which is formed by the above two straight lines. $\alpha, \beta$ indicates the angle between the movement direction of aircraft $A$ and the relative velocity vector and relative position vector. $\gamma$ indicates the angle between the edge of the protection area and the line connecting the two aircraft. If part of the protection area of aircraft $A$ overlaps the flight corridor of aircraft $B$, it indicates a potential conflict will occur [19].

$$\alpha = \arctan \frac{v_b \sin \varphi_r}{v_b \cos \varphi_r - v_a} \tag{5}$$

$$\beta = \left| \varphi_a - \arctan \frac{y_b - y_a}{x_b - x_a} \right| \tag{6}$$

Obviously, the conditions for avoiding conflicts are

$$|\alpha - \beta| \geq \gamma \tag{7}$$

$$\gamma = \arcsin\left(\frac{d_{min}}{\parallel z_r \parallel}\right) \tag{8}$$

## 4. Game Theory

### 4.1. Basics of Game Theory

Game theory originated from ancient realities of Go, games, and warfare, where players play games in which they consider how to act to maximize their own interests as well as considering the possible actions of other players. Game theory is, therefore, the study of how to choose the most sensible strategy to increase own profitability in a game where behavior interacts with each other.

Definition: In specific environmental conditions, the process by which several independent individuals, groups, or other organizations under certain constraints, choose and implement appropriate actions from their respective sets of strategies, either once or more, simultaneously, or sequentially, according to the information available, each gaining a corresponding benefit or outcome [20,21].

From the above definition, it can be concluded that a standard game should include the following six elements:

- Player: a decision subject who independently chooses to act in the process of a game.
- Information: the knowledge possessed by a player in the process of a game that is useful for decision making, mainly including the rules of the system and the decisions of other players.
- Order: the order in which the game parties make their decisions.
- Strategies: the entire set of behaviors or strategies that game parties can choose.
- Payoff: the gain or loss that results from a decision made by a game party.
- Outcome: The set of elements that interest the gamer, including the chosen strategy, payoffs, and strategic paths.

Both game theory and optimization theory can be used to increase individual gains or overall system gains by developing reasonable strategies, but the two methods apply to different scenarios. In optimization theory, all the decision variables affecting the outcome are held by a single intelligence, either a single player in the system or a single intelligence outside the system, which uses each player's behavior as a variable, constructs a multivariate optimization problem and solves it, and finally assigns the optimized variables to specific players and executes them to achieve an optimal solution to the system's payoff. In game theory, on the other hand, the decision variables that affect the outcome are held by all players, and each player considers the impact of the other players' strategies on their own payoffs when making decisions. Optimization theory is, therefore, suitable for centralized management systems, while game theory is suitable for distributed systems [20].

In a centralized air traffic control system, a conflict or potential conflict occurs between multi-aircraft, and the air traffic controller develops a conflict resolution strategy for each aircraft and executes it by each aircraft, a process that can be seen as an optimization process where the optimization variables are the control laws of each aircraft, and these decision variables are derived by the controller based on control experience, making conflict resolution less efficient. In the future distributed air traffic control system, multi-aircraft conflict is a game process; for all aircraft involved, the strategy space of each aircraft is the control law of that aircraft, and each strategy corresponds to a different gain, so the game parties will choose the corresponding strategy according to the maximum personal gain or the maximum system gain.

### 4.2. Horizontal Cross-Conflict Scenario Model Based on Cooperative Game Theory

Game theory considers the parties to a game to be rational, which is the objective of the decision-making behavior. The goal of maximizing individual interests is called "individual rationality", while the goal of maximizing group interests is called "collective rationality". Games can be divided into non-cooperative and cooperative games, depending on whether

there is a binding agreement between the parties to the game. In a non-cooperative game, each party is 'individually rational'; in a cooperative game, each party is 'collectively rational'.

In a non-cooperative game, the parties to the game reach a Nash equilibrium and are unable to increase their own gains by unilaterally changing their decisions, at which point they maximize their individual interests, but at this point, it is often not Pareto optimal for the system.

Safety and efficiency are the primary objectives of civil aviation. Individual aircraft in operation cannot be tolerated to maximize their own interests resulting in a reduction of system operation efficiency and safety; therefore, a coalition of aircraft involved in the conflict is considered.

Assuming that an aircraft detects a potential separation loss on the route, involving a total of $n$ aircraft, the coalition is $L = \{l_i \mid i \in [1, n]\}$; the strategy space of the system is $S = \{S_i \mid i \in [1, n]\}$, where $S_i$ is the strategy space of the $i^{th}$ aircraft, $S_i = \{s_{ij} \mid j \in [1, m]\}$, where $s_{ij}$ represents the $j^{th}$ strategy of the $i^{th}$ aircraft; and the benefits of the system $U = \{U_i \mid i \in [1, n]\}$, where $U_i$ is the benefits of the $i^{th}$ aircraft, $U_i = \{u_{ij} \mid j \in [1, m]\}$, where $u_{ij}$ represents the benefits of the $j^{th}$ strategy chosen by the $i^{th}$ aircraft. Then the welfare function of the coalition is

$$W = \sum_i k_i u_{ij} \tag{9}$$

where $k_i$ is the weight value and represents the priority of the $i^{th}$ aircraft.

The key elements of a cooperative game model in a horizontal cross-conflict scenario are as follows:

1.  Strategy space:

The strategy space is the set of all actions or strategies that the game parties can choose. The horizontal cross-conflict scenario only considers the game parties to accomplish the task of separation assurance by heading change. From the perspective of operational safety, the range of aircraft yaw angle is specified as $[-30°, 30°]$, where every $5°$ is a strategy, so the strategy space of aircraft $S_i$ is

$$S_i = \left\{ -\frac{\pi}{6} + \frac{(j-1) \times \pi}{36} \mid j \in [1, 13] \right\} \tag{10}$$

2.  Utility function:

The effectiveness function is the payoff corresponding to the strategy in the strategy space (Payoff). The payoff of aircraft maneuver flight is mainly related to yaw angle, flight time, flight path length, and fuel consumption considering that own ship only maneuvers by changing headings, so the path length and flight time of aircraft maneuver flight are linearly related, as shown in Equation (11)

$$dist_{\text{total}} = v_{\text{cruise}} \cdot t_{\text{total}} \tag{11}$$

where $dist_{\text{total}}$ is the flight path length, $v_{\text{cruise}}$ is the speed of the aircraft during the maneuver flight and $t_{\text{total}}$ is the maneuver flight time.

Because the speed and altitude are constant during maneuver flight, the fuel consumption of the aircraft can be linearly related to the flight time, as shown in Equation (12)

$$Q_{\text{total}} = f \cdot t_{\text{total}} \tag{12}$$

where $Q_{\text{total}}$ is the fuel consumption during the maneuver flight and $f$ is the fuel flow rate.

The gain during the maneuver flight is mainly related to the flight time and yaw angle. The larger the yaw angle, the smaller the benefit; the longer the maneuver time, the

smaller the benefit. Therefore, the utility function proposed in this paper for the horizontal cross-conflict scenario is

$$
\begin{cases}
u_{ij}(t_j, \theta_j) = (\mu_1, \mu_2) \times \begin{pmatrix} \cos \theta_j \\ -\lambda_i t_j \end{pmatrix} \\
\mu_1 + \mu_2 = 1
\end{cases}
\tag{13}
$$

In Equation (11), since $\cos \theta_j \in \left[ \frac{\sqrt{3}}{2}, 1 \right]$, and the magnitude of $t$ is usually 102 or 103, the adjustment parameters $\lambda_i$ are added to the utility function, $\mu_1$ represents the weight of the yaw angle contribution to the return and $\mu_2$ represents the weight of the maneuver flight time contribution to the return.

3.　　Priority:

Own ship in normal operation can treat designated traffic equally after a conflict, however, in actual operation there may be several factors that lead to different priorities of aircraft: (1) heavy aircraft in maneuver flight will consume more fuel than medium aircraft; (2) certain aircraft on special missions reduce large maneuvers as much as possible; (3) aircraft of a high delay rate need to minimize maneuver. Therefore, the priority of the aircraft involved in the conflict can be determined according to the classification of the aircraft, the class of the mission and the level of delay, thus ensuring that aircraft with a higher priority reduce the number and magnitude of maneuver flights.

4.　　Union welfare functions:

The coalition welfare function is the total benefit of the coalition. According to the Equations (9) and (13), the coalition welfare function can be obtained as

$$
W = \sum_i k_i \times (\mu_1 \cos \theta_i - \mu_2 \lambda t_i)
\tag{14}
$$

5.　　Conflict detection distance:

When the separation distance between own ship and designated traffic is less than the conflict detection distance, own ship uses the conflict detection algorithm to determine whether there is a potential separation loss between the own ship and the aircraft. A larger conflict detection distance will result in an increased computational load on own ship and unnecessary false alarms; a smaller conflict detection distance will result in an increased maneuver range for the aircraft and increased safety risks.

6.　　Safety separation:

The safety separation is the minimum separation that needs to be maintained between own ship and designated traffic in operation.

Literature [22] demonstrated that a single aircraft maneuvering at a large yaw angle to avoid a conflict is more costly and less beneficial than two aircraft avoiding at the same angle at the same time, this paper considers the more general case where the two aircraft fly at angular deflections of $\beta$ and $\varepsilon$ respectively to maintain separation.

Horizontal cross-conflict scenario as Figure 4 shown.

To facilitate modeling, take $AN/BN$ in Figure 4 as $y$-axis positive direction to establish the inertial coordinate system; the position coordinate of aircraft $A$ is $(x_A, y_A)$ and the vacuum velocity is $v_{A1}$, obtain the instantaneous wind field information $\overrightarrow{V_W}$ of the aircraft from the instantaneous wind field model. According to $\overrightarrow{V_{GS}} = \overrightarrow{V_{TAS}} + \overrightarrow{V_W}$ in the aircraft speed triangle, the ground speed $v_{A2}$ can be calculated, the nominal trajectory is $AO$ and the target point $A' = (x_{A'}, y_{A'})$; the position coordinates of aircraft $B$ is $(x_B, y_B)$ and the vacuum velocity is $v_{B1}$, the ground speed of the aircraft $v_{B2}$ is calculated, and the nominal trajectory is $BO$, the target point is $B' = (x_{B'}, y_{B'})$.

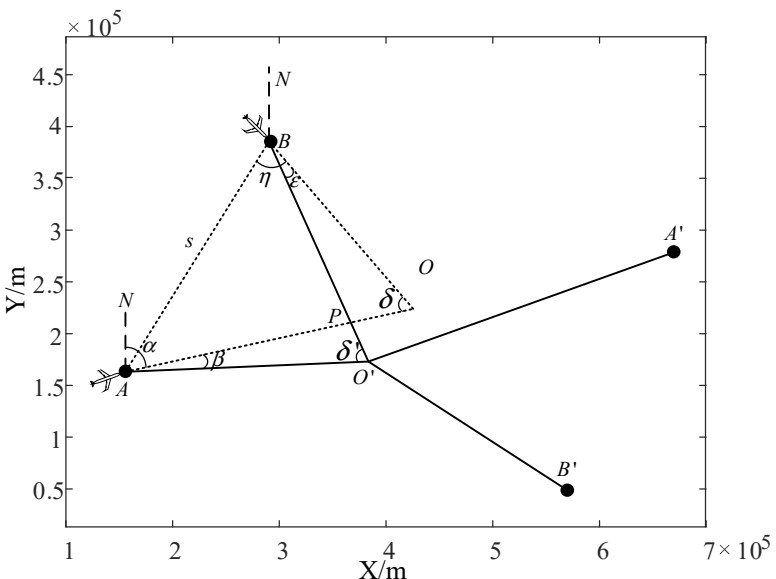

**Figure 4.** Horizontal cross-conflict scenario.

Assuming that the wind vector and ground speed remain constant throughout the conflict resolution process and the angle between the nominal trajectories of the two aircraft is $\delta$, the conflict detection algorithm detects that the two aircraft will clash in front of point $O$, and the priority of aircraft one and two are $k_1$ and $k_2$, respectively. Aircraft one and aircraft two adopt the strategy of flying with a deflection angle of $\beta$ and $\varepsilon$ respectively, $\beta \in S_1, \varepsilon \in S_2$, and the intersection of the new trajectory of them is $O'$, and the angle of the new trajectory can be deduced $\angle AO'B = \delta + \varepsilon - \beta$. Based on the relationship between the exterior angles of the triangle, we can obtain $\angle NBO' = \alpha + \varepsilon + \delta'$.

The linear equation of $AO'$ is

$$y - y_A = \cot(\alpha + \beta)(x - x_A) \tag{15}$$

The linear equation of $BO'$ is

$$y - y_B = \cot(\alpha + \varepsilon + \delta)(x - x_B) \tag{16}$$

Combine Equations (15) and (16), use Wolfram Mathematica solves the coordinates of $O'$ as

$$\begin{cases} x_{O'} = \dfrac{y_A - y_B - x_A \cot(\alpha+\beta) + x_B \cot(\alpha+\varepsilon+\delta)}{\cot(\alpha+\varepsilon+\delta) - \cot(\alpha+\beta)} \\ y_{O'} = \dfrac{y_B \cot(\alpha+\beta) - y_A \cot(\alpha+\varepsilon+\delta) + (x_A - x_B) \cot(\alpha+\beta) \cot(\alpha+\varepsilon+\delta)}{\cot(\alpha+\beta) - \cot(\alpha+\varepsilon+\delta)} \end{cases} \tag{17}$$

So, the lengths of $AO'$ and $BO'$ can be obtained as

$$L_{AO'} = \sqrt{(x_A - x_{O'})^2 + (y_A - y_{O'})^2} \tag{18}$$

$$L_{BO'} = \sqrt{(x_B - x_{O'})^2 + (y_B - y_{O'})^2} \tag{19}$$

The time $t_A$ required for aircraft one to travel from point $A$ to point $O'$ is

$$t_A = \frac{L_{AO'}}{v_A} \tag{20}$$

The time $t_B$ required for aircraft two to travel from point $B$ to point $O'$ is

$$t_B = \frac{L_{BO'}}{v_B} \tag{21}$$

In order to simplify matters, this chapter stipulates that aircraft one arrives at $O'$ while aircraft two is still flying on the $BO'$ sector, i.e., aircraft one is the first to fly over $O'$.

From Equation (16), the direction vector of the line $BO'$ is $(1, \cot(\alpha + \varepsilon + \delta))$, so that the coordinates of the position of aircraft two after the time of $t_A$ is $(x_B, y_B) + v_B t_A \left( \frac{1}{\sqrt{1+\cot^2(\alpha+\varepsilon+\delta)}}, \frac{\cot(\alpha+\varepsilon+\delta)}{\sqrt{1+\cot^2(\alpha+\varepsilon+\delta)}} \right)$ i.e., $\left( x_B + \frac{v_B t_A}{\sqrt{1+\cot^2(\alpha+\varepsilon+\delta)}}, y_B + \frac{v_B t_A \times \cot(\alpha+\varepsilon+\delta)}{\sqrt{1+\cot^2(\alpha+\varepsilon+\delta)}} \right)$.

After time $t_A$, aircraft one turns to fly straight to the target point $A'$, the direction vector of the straight line $O'A'$ is $(x_{A'} - x_{O'}, y_{A'} - y_{O'})$, currently aircraft two is still on the straight line $BO'$ towards the point $O'$.after $\Delta t$, the separation between them is minimal, at which point the position $P_A$ of aircraft one is

$$P_A = \left( x_{O'} + \frac{v_A \Delta t \times (x_{A'} - x_{O'})}{\| O'A' \|_2}, y_{O'} + \frac{v_A \Delta t \times (y_{A'} - y_{O'})}{\| O'A' \|_2} \right) \tag{22}$$

The position of aircraft two is

$$P_B = \left( x_B + \frac{v_B \times (t_A + \Delta t)}{\sqrt{1 + \cot^2(\alpha + \varepsilon + \delta)}}, y_B + \frac{v_B \times (t_A + \Delta t) \times \cot(\alpha + \varepsilon + \delta)}{\sqrt{1 + \cot^2(\alpha + \varepsilon + \delta)}} \right) \tag{23}$$

Then the position vector difference of the two aircraft is

$$\Delta P = P_A - P_B \tag{24}$$

The velocity vector $v_A$ of aircraft one is

$$v_A = v_A \times \left( \frac{x_{A'} - x_{O'}}{\| O'A' \|_2}, \frac{y_{A'} - y_{O'}}{\| O'A' \|_2} \right) \tag{25}$$

The velocity vector $v_B$ of aircraft two is

$$v_B = v_B \times \left( \frac{1}{\sqrt{1 + \cot^2(\alpha + \varepsilon + \delta)}}, \frac{\cot(\alpha + \varepsilon + \delta)}{\sqrt{1 + \cot^2(\alpha + \varepsilon + \delta)}} \right) \tag{26}$$

Then the velocity vector $\Delta v$ difference between the two aircraft is

$$\Delta v = v_A - v_B \tag{27}$$

When the separation between two aircrafts is the smallest, there is

$$\Delta P \cdot \Delta v^{\mathrm{T}} = 0 \tag{28}$$

The Equations (24), (27) and (28) can be combined to solve out the smallest separations $\Delta t$ and $d_{min}$.

The total time of maneuver flight of aircraft one is

$$t_1 = \frac{\| AO' \|_2 + \| O'A' \|_2}{v_A} \tag{29}$$

The total time of maneuver flight of aircraft two is

$$t_2 = \frac{\| BO' \|_2 + \| O'B' \|_2}{v_B} \tag{30}$$

Substituting Equations (29) and (30) into the coalition welfare function, Equation (14) yields the specific expression for the coalition welfare function.

$$W = k_1 \times (\mu_1 \cos \beta - \mu_2 \lambda \times \frac{\|AO'\|_2 + \|O'A'\|_2}{v_A}) + k_2 \times (\mu_1 \cos \varepsilon$$
$$-\mu_2 \lambda \times \frac{\|BO'\|_2 + \|O'B'\|_2}{v_B}) \tag{31}$$

By iterating through all the strategies in the strategy space, find $\beta$ and $\varepsilon$ that both satisfy the safe separation and maximize the coalition welfare, i.e., the separation assurance strategy for aircraft one and aircraft two.

## 5. Simulation of Horizontal Cross-Conflict Scenarios

### 5.1. Horizontal Cross-Conflict Scenario

Give the following horizontal cross-conflict scenario:

Suppose there are two aircraft in the control area, aircraft $A$ and aircraft $B$, which are flying at the same altitude. The initial position relationship of the two aircraft in the inertial coordinate system is shown in Figure 5. The instantaneous wind field information at the altitude of the two aircraft is obtained from Figure 2. The initial position of aircraft $A$ is (100, 100), the heading angle is 70°, and the ground speed is 800 km/h; The initial position of aircraft $B$ is (150, 400), the heading angle is 130°, and the ground speed is 700 km/h. the coordinate of the intersection $O$ is (400, 250), the coordinates of $A'$ and $B'$ are (700, 400), (650, 100) (in kilometers).

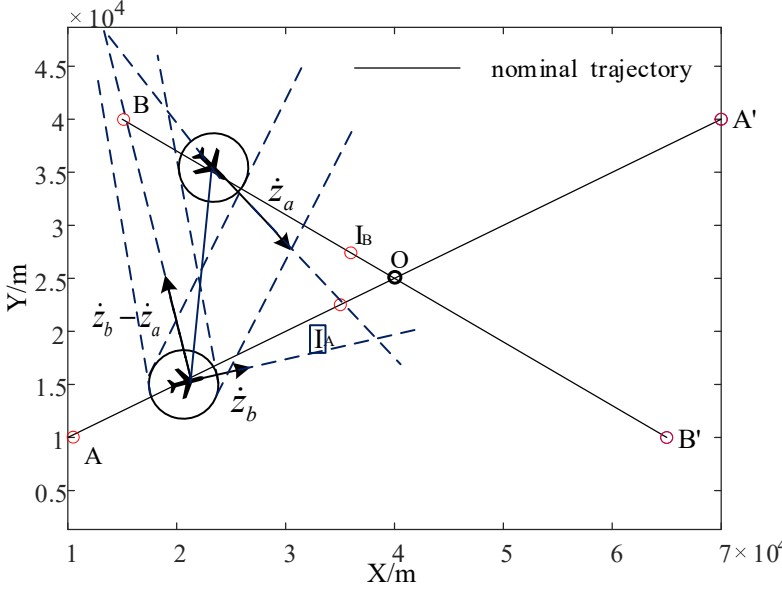

**Figure 5.** Conflict Detection Diagram of aircraft $A$ and aircraft $B$.

It can be seen from Figure 5 that the two aircraft are gradually approaching. According to the civil aviation air traffic management rules, the minimum safety separation is set to 10 km. The yaw angle range during separation assurance is $[-30°, 30°]$. This article assumes that the wind vector and the ground speed of the aircraft do not change when they are executing a separation assurance strategy.

Using the conflict detection algorithm, according to Formulas (5)~(8), we can obtain:

$$\alpha = \arctan\{700 \sin(60°) / [700 \cos(60°) - 800]\} = 57.46° \tag{32}$$

$$\beta = \left| 70° - \arctan(\frac{300}{50}) \right| = 73.87° \tag{33}$$

$$\delta = \arcsin(10/30) = 19.53° \tag{34}$$

$|\alpha - \beta| \geq \gamma$; therefore, it is determined that there is a potential conflict between them. Values of key parameters in the horizontal cross-conflict model:

1.  Utility function:

The utility functions of aircraft one and two are shown in Equation (13), $\lambda_1 = \lambda_2 = 0.01$. According to the different values of $\mu_1$ and $\mu_2$, the strategies of the alliance can be classified into minimum yaw angle strategy, minimum maneuver time strategy, and integrated optimal strategy.

The values of $\mu_1$ and $\mu_2$ for each of the three strategies are as follows:

Minimum yaw angle strategy: $\mu_1 = 1$, $\mu_2 = 0$.

Minimum maneuver time strategy: $\mu_1 = 0$, $\mu_2 = 1$.

Combined optimal strategy: $\mu_1 = 0.5$, $\mu_2 = 0.5$.

2.  Priority

The priority represents the importance of the parties in the game. In this paper, three combinations of priority are used: $k_1 : k_2 = 1 : 1$, $k_1 : k_2 = 2 : 1$ and $k_1 : k_2 = 1 : 2$ to simulate and observe the effect of priority on the game outcome.

### 5.2. Comparison of Three Separation Assurance Strategies

When the two aircraft have the same priority $k_1 : k_2 = 1 : 1$, the trajectory figures of the simulation results for the minimum yaw angle strategy, the minimum maneuver time strategy, and the combined optimal strategy are shown in Figures 6–8.

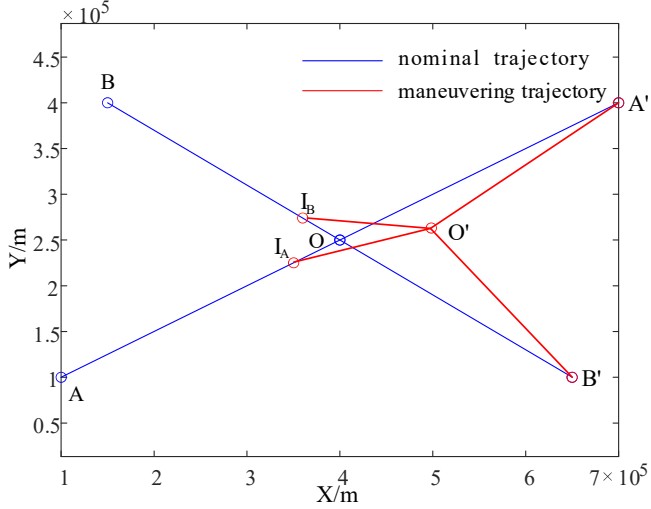

**Figure 6.** Trajectory diagram of minimum yaw separation assurance strategy.

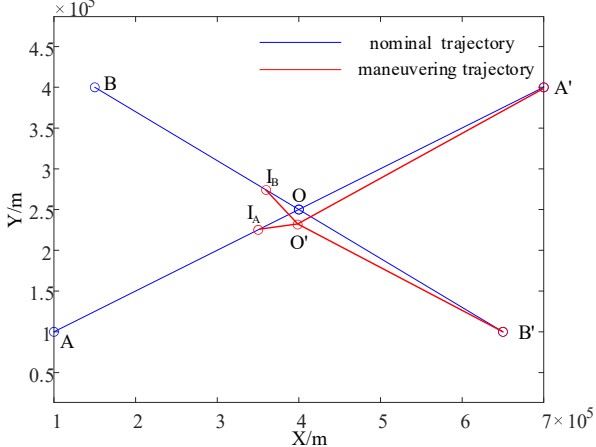

**Figure 7.** Trajectory diagram of minimum maneuver time separation assurance strategy.

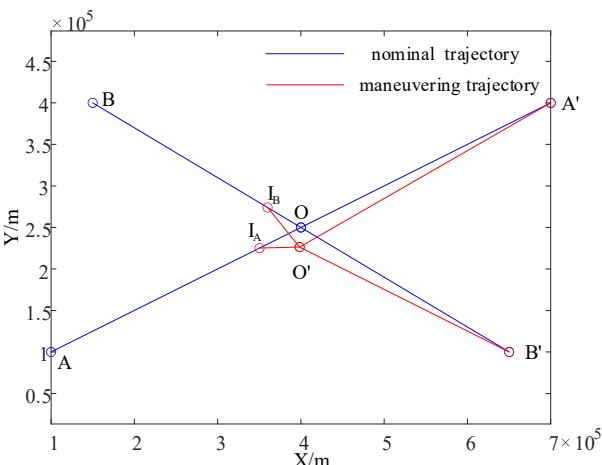

**Figure 8.** Trajectory diagram of integrated optimal separation assurance strategy.

When both aircraft have the same priority, key parameters for the three strategies during maneuver flight are shown in Table 3.

**Table 3.** Results of three separation assurance strategies.

| Separation Assurance Strategy | Aircraft one Priority: Aircraft Two Priority | $d_{min}$/km | $t_1$/s | $t_2$/s | $\beta$/$^o$ | $\varepsilon$/$^o$ |
|---|---|---|---|---|---|---|
| minimum yaw angle | 1:1 | 16.19 | 1876.54 | 1807.27 | 25 | −5 |
| minimum maneuver time | 1:1 | 15.84 | 1782.61 | 1767.26 | 20 | 15 |
| integrated optimal | 1:1 | 15.74 | 1785.98 | 1771.81 | 25 | 20 |

Where $d_{min}$ is the minimum horizontal separation between them during maneuver flight, $t_1$ is the total maneuver time of aircraft one, $t_2$ is the total maneuver time of aircraft two, $\beta$ is the yaw angle for aircraft one and $\varepsilon$ is the yaw angle for aircraft two.

*5.3. Effect of Priority on the Separation Assurance Control Law*

This section uses the minimum maneuver time strategy to calculate the autonomous separation assurance control law, according to $k_1 : k_2 = 1 : 1$, $k_1 : k_2 = 2 : 1$ and $k_1 : k_2 = 1 : 2$ respectively, three priority combinations for simulation; the simulation results of the three combinations are shown in Figures 7, 9 and 10.

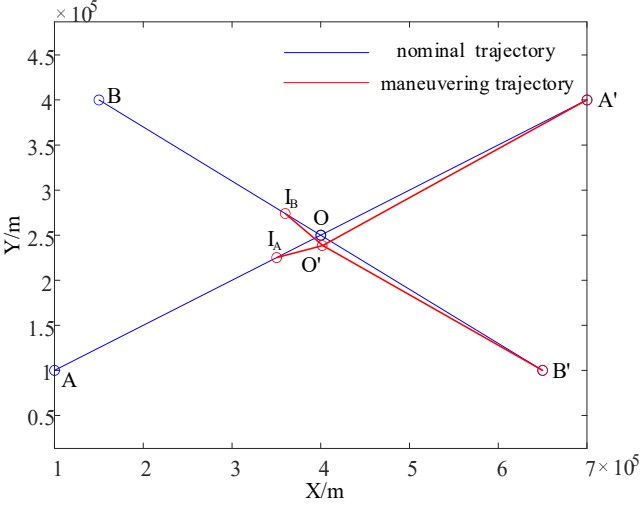

**Figure 9.** Trajectory diagram of maneuver separation assurance strategy at priority 2:1.

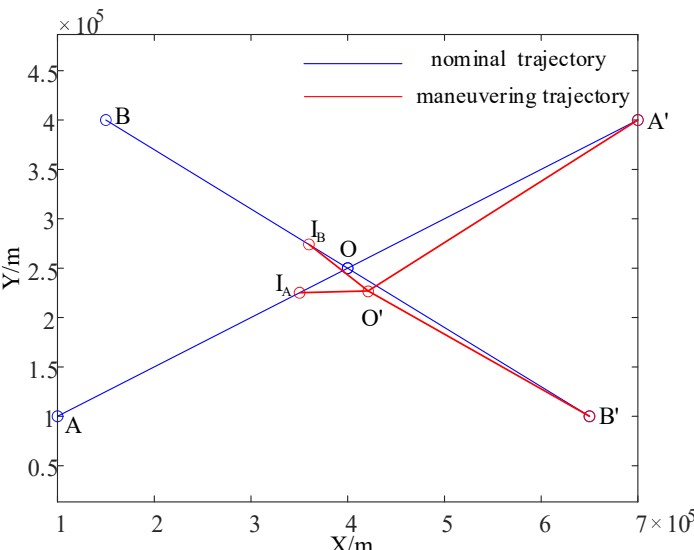

**Figure 10.** Trajectory diagram of maneuver separation assurance strategy at priority 1:2.

At different priorities, based on the minimum maneuver time strategy during maneuver flight, minimum horizontal separation $d_{min}$, total maneuver flight time $t_1$ of aircraft one, total maneuver flight time $t_2$ of aircraft two, yaw angle $\beta$, $\varepsilon$ of aircraft one and aircraft two are shown in Table 4.

**Table 4.** Simulation results at different priorities.

| Separation Assurance Strategy | Aircraft One Priority: Aircraft two Priority | $d_{min}$/km | $t_1$/s | $t_2$/s | $\beta/^o$ | $\varepsilon/^o$ |
|---|---|---|---|---|---|---|
| minimum maneuver time | 1:1 | 15.84 | 1782.61 | 1767.26 | 20 | 15 |
| minimum maneuver time | 2:1 | 16.01 | 1759.64 | 1771.84 | 15 | 25 |
| minimum maneuver time | 1:2 | 16.48 | 1821.17 | 1762.59 | 25 | 10 |

It can be seen from Table 4 that aircraft with higher priority tend to have smaller yaw angles and shorter maneuver flight time.

*5.4. Comparison Experiments*

In order to prove the effectiveness of the two-aircraft separation assurance strategy based on cooperative game theory, given the same horizontal cross-conflict scenario and constraints, with aircraft two as own ship, conduct own ship maneuver separation assurance experiment.

By comparing the two-aircraft separation assurance strategies when they have the same priority in Section 5.3, it can be found that although the minimum yaw angle strategy can minimize the sum of the absolute values of the yaw angles of the two aircraft, it may cause the aircraft to deviate from the air route farther, which in turn leads to an increase in the total maneuver flight time; while the minimum maneuver time strategy can make the aircraft fly over the conflict point earlier and resolve the conflict earlier. Therefore, the minimum maneuver time strategy is chosen to solve the separation assurance control law for a centralized, own ship maneuver separation assurance.

The position diagram obtained from the simulation experiment of the own ship maneuver separation assurance strategy is shown in Figure 11. Aircraft one and two follow the nominal trajectories of AA′ and BB′, respectively, and when aircraft one reaches the point $I_A$ and aircraft two reaches the point $I_B$, aircraft two uses a conflict detection algorithm and discover a potential separation loss from aircraft one. When aircraft one follows a predetermined trajectory without maneuvering, aircraft two calculates a maneuver trajectory which is the green one shown in Figure 11.

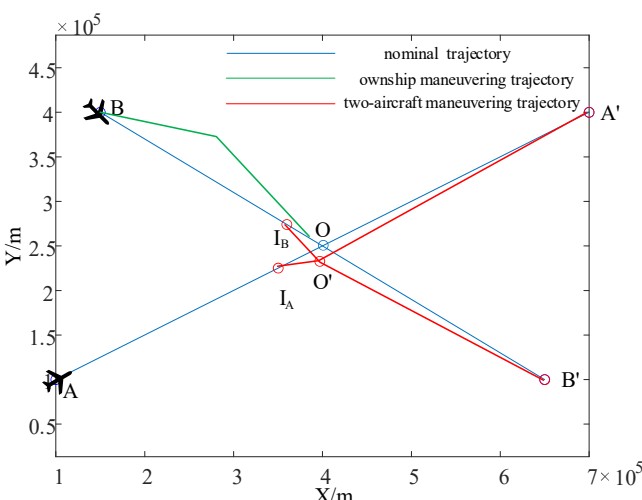

**Figure 11.** Comparison diagram of own ship and two-aircraft separation assurance strategy.

In the separation assurance process, the change of the two-aircraft separation of the two strategies with time is shown in the Figure 12.

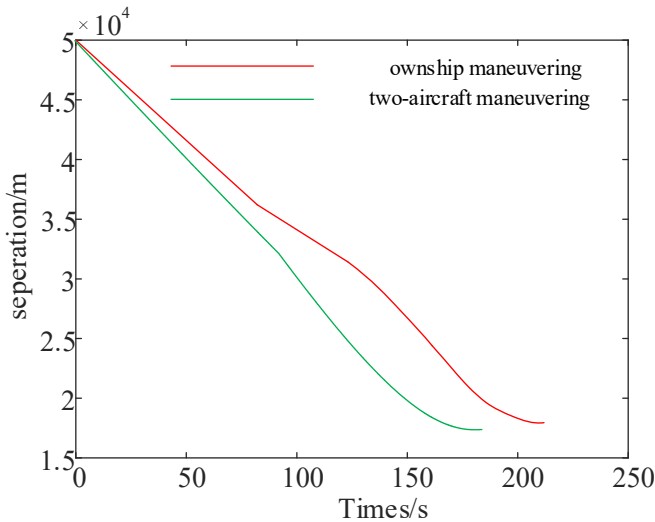

**Figure 12.** Separation-time diagram of own ship separation assurance strategy.

It can be seen from Figure 11 and Table 5 that the total maneuver time of the two-aircraft maneuver separation assurance strategy is 1782.61 s and 1767.26 s, respectively, and the yaw angles are 20° and 15°, respectively; The total maneuver time of own ship maneuver separation assurance strategy is 2279.64 s, and the yaw angle is 25°.

**Table 5.** Comparison experiment of the two air traffic management modes.

| Air Traffic Management Mode | Separation Assurance Strategy | $d_{min}$/km | $t_1$/s | $t_2$/s | $\beta$/° | $\varepsilon$/° |
| --- | --- | --- | --- | --- | --- | --- |
| Distributed-<br>two-aircraft cooperation | minimum maneuver time | 15.84 | 1782.61 | 1767.26 | 20 | 15 |
| Centralized-<br>own ship maneuver | minimum maneuver time | 16.07 | 0 | 2279.64 | 0 | 25 |

It can be seen from Figure 12 that under the condition of meeting the safety separation, in the 100~180 s conflict relief process, the two-aircraft maneuver separation assurance strategy not only takes much less time to resolve the conflict but also requires a much

smaller separation than that of the own ship. In summary, the two-aircraft maneuver requires less maneuver space and time in the separation assurance process.

Simulation results show that a two-aircraft maneuver separation assurance strategy based on cooperative game theory not only maneuvers at a smaller yaw angle but also takes much less time to resolve conflicts than a centralized, own ship maneuver separation assurance strategy.

## 6. Summary and Future Research

In this paper, we addressed the shortcomings of the own ship maneuver separation assurance by studying the two-aircraft autonomous separation assurance based on cooperative game theory; compared to the centralized one, the two-aircraft separation assurance strategy based on cooperative game theory jointly negotiated a suitable separation assurance control law, which avoids the separation loss without the need for significant maneuver flights and improves operational safety. However, due to the limitations of time and experimental conditions, the research work still had many shortcomings:

(1) The models and scenarios only considered two-aircraft and horizontal cross-conflict; our follow-up studies can be considered multiple aircraft and vertical cross-conflict.

(2) The wind vector and ground speed are constant in the process of conflict resolution, and the randomness of the wind vector should also be considered.

**Author Contributions:** Conceptualization, X.T., X.L. and P.Z.; methodology, X.T.; software, P.Z. and X.L.; validation, X.T., X.L. and P.Z.; formal analysis, X.T.; investigation, X.T.; resources, X.L.; data curation, X.T. and X.L.; writing—original draft preparation, X.T. and P.Z.; writing—review and editing, X.L. and P.Z.; visualization, X.T.; supervision, P.Z.; project administration, X.T.; funding acquisition, X.T. All authors have read and agreed to the published version of the manuscript.

**Funding:** This research was funded by the National Natural Science Foundation of China (61773202, 52072174) and the Foundation for National Key Laboratory of Science and Technology on Avionics System Integration (6142505180407) and the Civil Aviation Management Institute of China Key Foundation of General Aviation (CAMICKFJJ-2019-04).

**Institutional Review Board Statement:** Not applicable.

**Informed Consent Statement:** Not applicable.

**Data Availability Statement:** Not applicable.

**Conflicts of Interest:** The authors declare no conflict of interest.

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
