# Peer review of "Aircraft Autonomous Separation Assurance Based on Cooperative Game Theory"

_aerospace, doi:10.3390/aerospace9080421_

Round 1

Reviewer 1 Report

I found the article well-balanced and exciting. But as I read the title of this article, I thought about wind direction and speed versus height. The wind speed can reach 50km/h on the ground and increase with height significantly. Also, wind gusts are essential. The direction of the wind and wind gusts can change. Therefore, it should be considered in the proposed mathematical model because it looks simple to introduce. Moreover, the direction of wind gusts also should be analyzed.

The paper focuses on “4.1. Horizontal Cross-conflict Scenario”, but the solution is to change the height or time of arrival to conflict zones, so it should be visualized in figures. For example, trajectory lines can have gradient colors.

Small errors:

Figure 2 – the situation looks like a vertical conflict; please enter the coordinate system

Line 276  “Literature [Error! Reference source not found. demonstrated that a single aircraft “

Conclusion:

(1) A conflict detection algorithm based on the principle of ACAS was used to screen

the aircraft which would have potential conflict with own ship.

(2) using cooperative game theory, the aircraft in the conflict were involved as a coalition, a hybrid coalition welfare function consisting of yaw angle and manoeuver flight time was proposed, and the concept of priority was raised which was used to ensure that aircraft of higher priority made as few large manoeuver flights as possible.

(3) A cooperative game

Punctation (2) in this method sentence should start “Using…”

Reviewer 2 Report

This paper presents a method to solve horizontal cross-conflict between two aircraft using cooperative game theory for future air traffic management. The study is interesting and could be useful to improve future air traffic management. My comments to further improve the paper are as follows:

The conflict detection and resolution method used in the paper is simple in that it only uses one strategy to solve horizontal conflicts between two aircraft. Some sophisticated conflict detection and resolution methods (e.g., https://doi.org/10.1016/j.tre.2021.102407) for multiple aircraft could be used to improve the study. The authors might include some discussion of these methods as a future improvement of the paper.

Reviewer 3 Report

Manuscript ID: aerospace-1782093

Manuscript Title: Aircraft Autonomous Separation Assurance Based on Cooperative Game Theory

The following modifications are suggested:

1.      In the introduction section, p. 2, please assign the reference number for “Claire Tomlin et al.…..”, “Yang Min et al….” and “Cheng Ying et al…..” and any similar unassigned one.

2.      The novelty of the manuscript is not sufficiently clear, thus, rewrite it better.

3.      The number of papers reviewed in the introduction is few, please increase it.

4.      Simulation results should be compared with the literature to be more acceptable.

5.      The conclusions in Section 5 must be supported by the numbers obtained from the results.

6.      The first and second points in the conclusions section are closer to describing the work than to the conclusions, so please put them in the introduction of section 5 as an explanation or delete them.

7.      The number of references is relatively low. Therefore, add more (recent) references.

Round 2

Reviewer 1 Report

Dear authors I like upgraded version of publication.

Reviewer 3 Report

The authors have improved their manuscript. It can now be accepted in its current form.